# Early Onset Metastatic Colorectal Cancer: Current Insights and Clinical Management of a Rising Condition

**DOI:** 10.3390/cancers15133509

**Published:** 2023-07-05

**Authors:** Bianca Medici, Beatrice Riccò, Eugenia Caffari, Silvia Zaniboni, Massimiliano Salati, Andrea Spallanzani, Ingrid Garajovà, Stefania Benatti, Chiara Chiavelli, Massimo Dominici, Fabio Gelsomino

**Affiliations:** 1Department of Oncology and Hematology, Division of Oncology, University Hospital of Modena, 41124 Modena, Italy; bia.medici31@gmail.com (B.M.); beatrice.ricco@hotmail.it (B.R.); eugeniacaffari@gmail.com (E.C.); silvia.zaniboni24@gmail.com (S.Z.); salati.massimiliano@aou.mo.it (M.S.); spallanzani.andrea@aou.mo.it (A.S.); stefania.benatti@unimore.it (S.B.); massimo.dominici@unimore.it (M.D.); 2Medical Oncology Unit, University Hospital of Parma, 43100 Parma, Italy; ingegarajova@gmail.com; 3Laboratory of Cellular Therapy, Division of Oncology, Department of Medical and Surgical Sciences for Children & Adults, University of Modena and Reggio Emilia, 41121 Modena, Italy; chiara.chiavelli@unimore.it

**Keywords:** colorectal cancer, early onset, screening

## Abstract

**Simple Summary:**

In this review, the authors will discuss the highlights of early onset colon cancer (EOCRC), an increasing but still unclear phenomenon. Indeed, the risk factors are largely similar to classic colon cancer (CRC), but it is still unclear what causes an increased risk of CRC at an early age. In addition, the issue of screening, which allows early diagnosis of the disease, is also important; the younger segment of the population is not included in such programs except if they belong to the highest-risk groups. If the phenomenon of EOCRC becomes more prominent, screening programs will have to be reviewed but with consideration of the consequences for health economics. In this review, we will focus on the epidemiological, prognostic, clinico-pathological, molecular, and therapeutic features of EOCRC; in particular, we will try to highlight the characteristics of the metastatic stage and the differences with CRC in older segments of the population.

**Abstract:**

Despite a recent overall decrease in colorectal cancer (CRC) incidence and mortality, there has been a significant rise in CRC diagnoses in young adults. Early onset colorectal cancer (EOCRC) is defined as CRC diagnosed before the age of 50. Possible predisposing conditions include not only genetic syndromes but also other risk factors, such as microbiome alteration, antibiotic exposure, obesity, diabetes mellitus, and inflammatory bowel disease. EOCRC tends to be diagnosed later than in the older counterpart because of a lack of awareness and the fact that screening for CRC usually starts at the age of 50. Furthermore, CRC in young adults seems to be related to unique molecular features and more aggressive clinical behavior. This paper aims to provide an in-depth review of this poorly understood subject, with a comprehensive review of the state of the art and considerations for future perspectives.

## 1. Introduction

Early onset colorectal cancer (EOCRC) is defined as CRC diagnosed in individuals younger than 50, which is generally considered the ideal age to start screening programs in the average-risk population. Although the overall incidence of colorectal cancer (CRC) is declining, the number of new diagnoses in patients younger than 50 is alarmingly increasing [1,2]. Modifiable and nonmodifiable factors, such as antibiotic exposure, obesity, a Western diet, diabetes mellitus, inflammatory bowel disease (IBD), environmental pollution, and pesticide use, might be among the possible causes [3]. However, the exact reasons for this rising phenomenon are still unknown.

EOCRC seems to have different features than CRC in older patients. EOCRC generally develops with more aggressive features, is diagnosed at a more advanced stage [4,5,6,7], and has stronger metastatic potential [8]. On the other hand, young people with metastatic cancer have better overall survival (OS), probably related to better performance status, lower comorbidities, higher tolerance to chemotherapy treatments, and lower postsurgical mortality [9].

Of note, the advanced stage at diagnosis might be related to the fact that screening campaigns do not involve the population under the age of 50, except among individuals with a family history of CRC or those affected by chronic IBD [3,10]. The noteworthy increase in CRC cases among young people in the last decade might lead to considering the need to lower the age of starting screening; however, these measures would result in increased healthcare costs. 

Given the greater proportion of young patients that present with advanced disease at the time of diagnosis, young adults with Stage IV EOCRC represent a small but increasing and relevant population [11] that needs greater attention and further study. For this reason, we herein present a comprehensive review of the literature on EOCRC, with a particular focus on metastatic disease. 

## 2. Epidemiology

CRC is the third most common malignancy and cause of cancer death worldwide in both genders, mostly over the age of 50 [12]. Since the mid-1980s, incidence and mortality have each decreased, likely due to both the start of screening programs and the optimization of disease management [13]. However, this progress is confined to older individuals, and multiple studies have revealed an alarming increasing incidence among people younger than 50 [10,14,15,16]. Namely, in the last twenty years, the median age of CRC diagnosis has decreased from 72 to 66. In addition, 10 to 20% of CRC diagnoses involve people younger than 50, and about three-quarters of them are aged 40–49 (SEER Stat Database) [17]. Of note, the increase in EOCRC IRs has been mainly driven by rectal cancer diagnoses [18], which have risen by more than 90% from the beginning of the 1990s to 2016 (from 2.6 to 5.1/100,000) compared to an increase of about 40% for colon cancer [17]. With regard to sex differences, whereas the incidence of CRC in the 55–74 age group is almost 50% greater in men than in women, it is comparable between men and women diagnosed earlier than at 40 [19,20].

Considering the current data and despite general trends toward population aging, a retrospective cohort study foresees by 2030 an increase in colon cancer diagnosis of 90% in the 20–34 age cohort and 27.7% in the 35–49 age cohort, with an even higher rise for rectal cancer diagnosis of 124% and 46% for the two subgroups, respectively [21,22].

### 2.1. Geographic Differences

As displayed in Figure 1 and Figure 2, global EOCRC incident rates (IRs) fluctuate from 3.5 per 100,000 inhabitants in India to 12.9 in the Republic of Korea [23]. In the last decade, an increasing IR was recorded in 19 out of 36 countries, among which 9 (e.g., Australia, Germany, and the US) had stable or declining trends in older adults. Only three countries (Austria, Italy, and Lithuania) exhibited a decrease in EOCRC IRs [18]. A similar distribution emerged from a further recent study [24], which also showed how the increase is mainly attributable to rectal cancer, with the exception of the United Kingdom and Brazil. The highest incidence of EOCRC was found in females in Switzerland (4.2/100,000) and in males in the Republic of Korea (4.6/100,000), with no difference in trend variation between rectal and colon cancer [25].

In the United States (US), age-specific CRC risk has returned to the levels recorded in those born around 1890 [15]. Here, an increase in colon cancer IRs was seen both in the 20–39 age cohort (from 1% to 2.4% annually since the mid-1980s) and in the 40–54 age cohort (from 0.5% to 1.3% annually since the mid-1990s). A faster increase in rectal cancer IRs (3.2% annually from 1974 to 2013 in adults aged 20–29 years) was also reported in the US, where from 1990 to 2013, the number of diagnoses in the under-55 population doubled from 14.6% to 29.2% [15]. It is noteworthy that in the US, ethnic differences have been reported; in particular, non-Hispanic black individuals have been reported to be at a higher risk of EOCRC development [22,23,24,25,26,27], especially in rural areas [28]. 

A similar trend was also observed in European countries, where from 2004 to 2016, CRC IRs increased by 7.9% annually in the age group of 20–29 years, 4.9% among those aged 30–39 years, and 1.6% in the 40–49 cohort [29].

With regard to non-Western populations, in recent years, the incidence of EOCRC has also been increasing in Arabic countries, which could be attributed to improved diagnostic strategies and changes in lifestyle and dietary habits, which have become more similar to those of Western countries [30]. An IR increase has also been reported in Iran and Egypt [19]. The age at CRC diagnosis in Africa and Asia is lower than in Europe and America [31], probably due to heritable causes, although no causes can be found in the literature to explain a different epidemiological trend from the Western population.

When talking about geographic differences, an important factor to consider is the presence of a private healthcare system, as EOCRC seems more prevalent among patients with no insurance coverage or ready access to care (16.5% vs. 4.7%), often of nonwhite ethnicity (29.5% vs. 17.6%) [7].
Figure 1Map showing EOCRC incidence rates worldwide. Red countries are those in which an increased incidence rate of EOCRC has been documented [18,19,20,31,32].
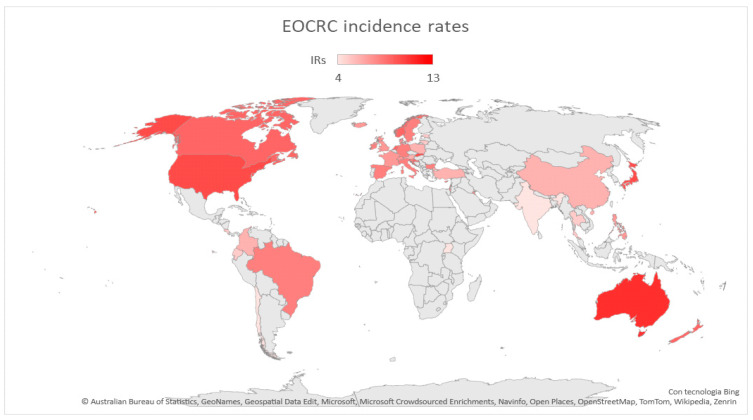

Figure 2Map showing EOCRC incidence annual per cent change (APC) in the last 30 years. Red countries are those in which an increased APC has been documented. Green countries have experienced a decrease in APC [18,19,20,31,32].
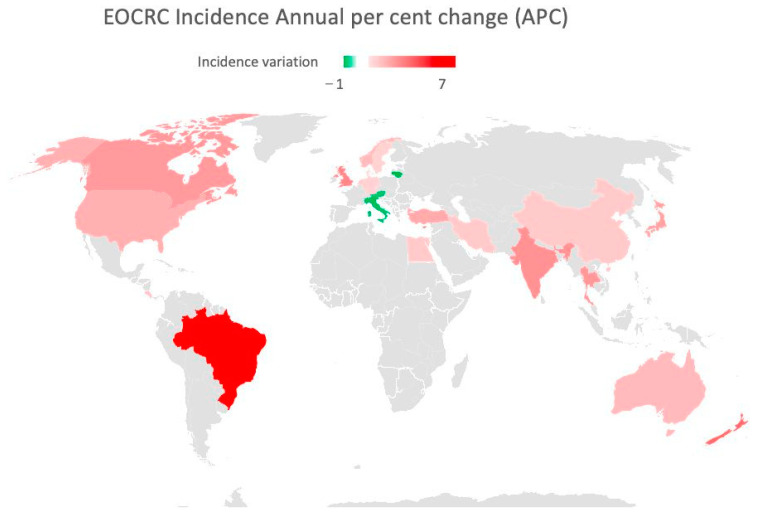



### 2.2. Screening Programs

Considering that most EOCRC diagnoses are made in people with an average screening risk, and almost half of these patients are aged between 45 and 49 (SEER-Stat Database), since 2018, the American Cancer Society has indicated 45 as the optimal age to initiate CRC screening [33].

Screening in individuals younger than 50 is universally recommended for people with an elevated risk of CRC because of chronic IBD, familial syndromes, or with a family history of CRC in a first-degree relative (FDR) [10]. In these cases, screening colonoscopy is indicated by the age of 40 or, for some international societies (e.g., the American College of Gastroenterology, U.S. Multi-Society Task Force of Colorectal Cancer), 10 years prior to the age of diagnosis of advanced adenoma in the FDR before the age of 60, with a follow-up colonoscopy every 5 years. Other high-risk groups that received conditional recommendations to initiate early screening include African American individuals [34], cystic fibrosis patients [32], and people who underwent pelvic radiation (>30 Gy) at a young age [35].

## 3. Risk Factors

As shown in Table 1, EOCRC might be caused by both hereditary and nonhereditary conditions. In this section, we provide an overview of the risk factors for EOCRC development which have been identified so far.

### 3.1. Hereditary EOCRC

EOCRC is mainly a sporadic disease (80%): only about 1 out of 5 CRC patients diagnosed under the age of 50 carry a germline mutation, and almost half of them show no family history related to the corresponding hereditary cancer syndrome [36]. However, considering that hereditary cancer syndromes account for 2–5% of overall CRC diagnoses, the rate in the younger population is significantly higher [20]. Patients with a CRC diagnosis under the age of 35 should receive genetic counseling [37]. The 2022 National Comprehensive Cancer Network (NCCN) guidelines recommend that all patients with CRC, regardless of age or family history, should undergo multigene panel testing.

#### 3.1.1. Hereditary Nonpolyposis CRC or Lynch Syndrome (LS)

LS is the most common cause of hereditary CRC and accounts for 2–3% of all CRC cases [36,38]. It is associated with germline mutations in the DNA mismatch repair (MMR) genes (MLH1, MSH2, MSH6, and PMS2) or epithelial cell adhesion molecule (EPCAM). Patients with LS have a 70% lifetime risk of developing CRC, with a 40% chance of diagnosis before the age of 40 [20].

#### 3.1.2. Adenomatous Polyposis Syndromes (APSs)

APSs include familial adenomatous polyposis (FAP), polymerase proofreading-associated polyposis (PAPP), NTHL1-associated polyposis (NAP), and MUTYH-associated polyposis (MAP). FAP is caused by germline mutations in the APC tumor suppressor gene, and it has been associated with 1% of all CRC cases. 

In the process leading from adenoma to tumor, there is an initial mutation of the APC gene, which normally inhibits cell proliferation, resulting in the instability of other genes, such as KRAS, SMAD4, and P53 [39]. Therefore, APC mutations promote the translocation of beta-catenin into the nucleus with the subsequent transcription of certain proto-oncogenes, such as c-myc and cyclin D1, which play a central role in the carcinogenesis process. APC normally results in a decrease in the beta-catenin levels through the phosphorylation of GSK3-beta. Beta-catenin is involved in the Wnt-1 signaling pathway: it binds Wnt-1, which causes an inhibition of GSK-beta and, thus, an increase in the beta-catenin levels [40]. Mutation of the APC gene results in imbalances in the Wnt pathway and hence in cell cycle processes [41].

Carriers of the APC mutation typically develop classic colonic polyposis with 100 to 1000 adenomas and show close to a 100% lifetime risk for CRC. The presence of attenuated phenotypes (20–100 polyps) is acknowledged [36,42]. The median age of CRC diagnosis is 39 years old; therefore, screening using sigmoidoscopy or colonoscopy every 1 to 2 years is recommended from the age of 10 [20,43].

#### 3.1.3. Other Germline Alterations Associated with EOCRC

Li–Fraumeni syndrome, associated with germline TP53 mutations, causes less than 1% of EOCRC. Several other genes have been explored (e.g., BRCA1-2, CHEK2, SMAD4, ATM, BRF1, and BUB1-2); however, their association with EOCRC has been demonstrated in only 2% of cases [20].

#### 3.1.4. Cystic Fibrosis (CF)

CF has recently been listed as a familial CRC syndrome. The loss of CFTR alters a variety of cellular processes and might lead to carcinogenesis. By the age of 40, 50% of CF patients develop adenomas, among which 25% are aggressive advanced adenomas or adenocarcinomas. This led to the development of new CF CRC screening recommendations, and the initiation of endoscopic screening has been lowered to the age of 40 in CF patients and to the age of 30 in organ transplant recipients [32,44].

### 3.2. Sporadic EOCRC

The rising incidence of EOCRC has involved patients born after 1960, which suggests population-level behavioral changes or earlier points to exposure [22]. We herein present a list of modifiable risk factors that appear to have a strong impact on EOCRC development.

#### 3.2.1. Metabolic Syndrome

Compared with individuals without metabolic syndrome, those with 1, 2, or ≥3 metabolic comorbidities have a 9%, 12%, and 31% higher risk of EOCRC, respectively. These positive associations are true for colon but not rectal cancer [45].

Obesity is surely one of the main modifiable risk factors associated with CRC. A prospective study involving US female nurses aged 25 to 42 revealed that the relative risk (RR) for incident EOCRC for each 5-unit increment in BMI in women was 1.2, considering weight gain since early adulthood [46]. Several meta-analyses demonstrate similar findings [47,48,49]. In contrast, a case-control study conducted among US veterans displayed a correlation between low body weight and an increased risk of EOCRC. Nevertheless, this was primarily due to the fact that the BMI assessment was conducted prior to the diagnostic colonoscopy rather than in early adulthood [50]. Among the reasons abdominal obesity is considered a cancer-related risk factor is that increased adipocyte tissue generates adipokines implicated in carcinogenesis [16].

Type II diabetes mellitus (DM) is also associated with an increased risk of CRC, with a RR of 1.3. DM individuals might develop CRC 5 years earlier than the average population due to prolonged exposure to hyperinsulinemia, which is directly implicated in oxidative stress and in tumor proliferation promotion [16,22,51].

Interestingly, even though metabolic syndrome mostly affects people over 50, CRC incidence in this age group is stable or declining. This might be due to the fact that older individuals are more often on maintenance medications (e.g., aspirin, statins, and metformin), which may have a chemopreventive role. For example, it has been reported that low-dose metformin was associated with a decrease in metachronous adenomas or polyps after a polypectomy [52,53].

#### 3.2.2. Diet

High consumption of red and/or processed meat and sugary beverages is associated with an increased risk of CRC, whereas vegetables, fruits, fiber, and calcium are related to a lower CRC risk [16]. There is also evidence that a sulfur microbial diet is associated with an increased risk of advanced adenoma (but not serrated lesion) formation in those under the age of 50. This supports a role for dietary interactions with gut sulfur-metabolizing bacteria in EOCRC carcinogenesis, possibly beginning in adolescence [54]. It is noteworthy that, as the carcinogenic potential of diet likely starts decades prior to CRC presentation, it is difficult to interpret its influence when we only know self-reported diet-related information near the time of diagnosis.

#### 3.2.3. Physical Activity

Some meta-analyses [55,56] among all age groups have found a positive correlation between physical activity and a reduced risk of CRC, with a stronger association for colon cancer than for rectal cancer. This might be attributed to several mechanisms, such as lower BMI, insulin levels, and inflammation factors [22]. Sedentary behavior conferred an increased risk of CRC. In fact, prolonged exposure to fecal carcinogens, such as secondary bile acids, in sedentary individuals compromises glucose homeostasis and decreases vitamin D levels. Conversely, an active lifestyle can improve blood flow and muscle contraction and lead to improved glucose regulation and endothelial function [2].

#### 3.2.4. Alcohol and Tobacco Use

Alcohol consumption and smoking are associated with a significant increased risk of CRC development (OR 1.25 for alcohol intake of more than 3 drinks per day; RR 1.71 for smokers) and EOCRC incidence (RR 1.71) [57].

#### 3.2.5. Aspirin and Nonsteroidal Anti-Inflammatory Drugs (NSAIDs)

Long-term use of NSAIDs and aspirin have displayed a protective role against the development of EOCRC [50]. The American College of Gastroenterology (ACG) guidelines provide a conditional recommendation for aspirin use for CRC prevention in patients aged 50–64 [16].

#### 3.2.6. Chronic Inflammation

IBD promotes carcinogenesis in CRC as it leads to an inflammatory environment, with increased free radical species, circulating adipokines, and upregulated antiapoptotic and proliferative pathways [16,58]. Of note, inflammation and immune responses might differ across races and ethnicities: a stronger lymphocytic reaction was found in tumors of African Americans, which may be associated with increased CRC risk in this population [22].

#### 3.2.7. Intestinal Dysbiosis

Intestinal dysbiosis can be caused by various factors, such as high BMI, dietary habits, and chronic inflammation. Specific intestinal bacterial species (e.g., *Bacteroides fragilis*, *Fusobacterium nucleatum*, *Streptococcus bovis*, and *Escherichia coli*) can promote CRC through the alteration of colonic integrity, bile acid composition, and the production of toxins and metabolic products of short chain fatty acids (SCFAs), which can influence systemic inflammatory responses [16]. Furthermore, a recent study reported a significant association between antibiotic use and colon cancer, particularly in the EOCRC cohort [59], which might be due to their effect on gut microbiota and the alteration of its composition [22]. Of note, the same study showed no correlation between antibiotic use and rectal cancer [59].

#### 3.2.8. Pollution

In recent years, several studies have tried to demonstrate a correlation between CRC and pollution [60]. Residing in the proximity of industries may be a risk factor for CRC, especially at a distance of 1–3 km and when pollutants are released in the air rather than in the water [61].

#### 3.2.9. Acute Infections

Epidemiological analyses have highlighted that individuals who experience at least three infections with fever above 38.5 °C have a 40% lower risk of cancer development. This might be due to the fact that a high body temperature leads to the killing of cancer cells and the release of cancer-associated antigens, which may enhance an antitumor immune response [2,62]. In recent decades, there has been a decreased incidence of infective episodes among younger people with higher antibiotic and antipyretic consumption and, thus, lower febrile events.

#### 3.2.10. Association between Modifiable Risk Factors and Anatomical Sites

Many studies have observed that risk factors differ according to the anatomical site [63,64].The reasons are not precisely defined, but they can be many. First of all, the proximal colon and the distal colon and rectum have a different embryological origin, and each anatomical site performs specific functions. Secondly, the microbiota and fecal composition, enzymes, and metabolites change along the intestinal tract [65]. Therefore, these elements might increase susceptibility to certain risk factors, influencing the process of carcinogenesis [66].

Murphy et al. recently published the data of a multinational cohort study, describing the association between lifestyles and the anatomical sites of CRC onset. They showed that obesity is a stronger risk factor for the development of cancer in the proximal (HR 1.31; 95% CI: 1.18–1.47) and distal colon (HR 1.32; 95% CI: 1.20–1.45) than in the rectum (HR 1.10; 95% CI: 1.01–1.20). Conversely, physical activity reduces the incidence of proximal (HR 0.74; 95% CI: 0.63–0.87) and distal colon cancer (HR 0.89; 95% CI: 0.76–1.05) but does not significantly affect rectal cancer. Smoking, on the other hand, appears to increase the risk of distal colon and rectal cancer (HR 1.27; 95% CI: 1.13–1.43 and HR 1.20; 95% CI: 1.09–1.33, respectively). Alcohol consumption, diabetes, and the use of NSAIDs generally increase the risk of CRC but do not appear to correlate with a specific anatomical site [63] (Figure 3).

## 4. Prognosis

While in many other cancer types, young people and adolescents have been shown to have a worse prognosis [67,68,69], the data regarding outcome and survival in metastatic EOCRC (mEOCRC) are conflicting and still under study. This controversy arises from differences in the study populations (e.g., different CRC stages) and small patient samples [70].

Hawk et al. collected data from 63.894 patients with mCRC throughout a period of 35 years and showed an improvement in OS for young patients compared to older patients (15 vs. 9 months, respectively) [71]. Other authors suggested that mEOCRC patients have similar survival outcomes compared to their older counterparts [70,72]. In contrast, many studies indicate that EOCRC displays a worse prognosis. Lieu et al. described a significant increase in the risk of disease progression (22%) compared to patients aged >50 years (15%) [73]. This indicates that mEOCRC biological features lead to a poorer response to the first line of treatment and faster disease progression. Rho YS et al. conducted a study on 498 mCRC patients and showed inferior PFS on first-line therapy for patients under 45 compared to older patients (HR 1.96; 95% CI 1.04–3.68) [74]. On the other side, older patients have an increased risk of death during the first years of treatment, owing to comorbidities and reduced tolerance to chemotherapy. Metastatic late-onset CRC (mLOCRC) has an estimated 42% increased risk of death compared to 19% for younger patients [73]. Other research confirmed that EOCRC has a poorer prognosis compared to LOCRC due to worse prognostic factors, such as high-grade tumors, poorly differentiated cells, and mucinous or signet cell histology [75,76]. Thereby, tumor biology seems to be a more important prognostic indicator for CRC than age [77]. 

In the literature, the discrepancies in mEOCRC outcomes can be influenced by the inclusion or exclusion of hereditary cancer syndromes in data collections. Indeed, Lynch syndrome-associated CRC typically shows increased survival rates rather than sporadic cases due to the possibility of responding to immunotherapy [5,78,79].

It is widely reported that young adults tend to receive more aggressive treatments in terms of both surgery and chemotherapy in comparison to older patients with the same disease stage [80,81]. For instance, metastasectomy or local ablative therapy are procedures more likely to be offered to younger patients with Stage IV CRC than their older counterparts, as they generally display a better performance status and fewer comorbidities [73,82]. However, this attitude does not significantly affect the outcome and survival of patients, possibly representing an overtreatment with increased toxicities [81]. 

Regarding mortality rates, Siegl et al. recently published updated CRC statistics from the American Cancer Society. The results show that in the last few years, death rates have an opposite trend based on age, similar to the incidence trend. Among adults older than 50 years old, mortality has had a steady decline of 0.5% annually since 2005 thanks to improved treatments. On the other hand, young adults have presented a constantly increasing death rate of approximately 1% per year since 2004 [83].

## 5. Clinicopathological Features

Several studies have shown that EOCRC has peculiar features compared to LOCRC, regarding its clinical presentation, anatomical sites, and histological classification.

EOCRC is typically diagnosed in more advanced pathologic stages (Stages III or IV) [84,85]. A study performed at the Stanford Cancer Institute detected that 72% of young people received a diagnosis of CRC at an advanced stage compared to 63% of older patients (*p*= 0.03) [86]. Moreover, in the US, 27% of EOCRC patients have distant metastases at the time of diagnosis compared to 20% of LOCRC patients [83]. The lack of screening programs is the first reason why early or accidental cancer detections are not made. Only people with hereditary cancer syndromes may periodically undergo diagnostics exams for early diagnosis. However, the number of early-onset sporadic cases is increasing, and the diagnosis is made only when symptoms occur [87]. 

The main symptoms observed are rectal bleeding, abdominal pain, and changes in bowel habits. It is reported that hematochezia can be developed from 6 months to 1–2 years before diagnosis [86,87,88]. Chen et al. showed that the time between the onset of symptoms and the diagnosis is 40% longer in young patients than in elderly patients (128 days [IQR 60–265] versus 79 days [IQR 31–184], *p* < 0.05) [86]. The first signs can be underestimated by patients and doctors due to the low clinical suspicion of oncological diseases in this kind of population. Clinicians tend to correlate the rectal bleeding or the abdominal pain to more common and benign pathologies, such as hemorrhoids or IBD, and they do not investigate immediately with specific exams to exclude cancer [89]. Furthermore, young people do not receive adequate information about warning symptoms, and they tend to delay a medical visit [10]. In order to anticipate diagnosis in young people, stool examination and/or colonoscopy may be appropriate to investigate rectal bleeding or changes in bowel habits [29,90].

Many studies have shown a different location of CRC based on age. The majority of mEOCRC is localized in the left colon (rectum, rectosigmoid colon, and distal colon), with a percentage of 41% compared to 34% of mLOCRC [29,86].

An increased predisposition to synchronous and metachronous tumors or finding polyps afterward is also reported [5].

Another explanation of this late diagnosis involves tumor biology. EOCRC tends to have histopathological features associated with rapid carcinogenesis and a tendency to metastasize precociously. Young patients’ cancer is usually poorly differentiated and high grade, and it shows perineural, lymphovascular, and venous invasion [91,92]. EOCRC is more frequently characterized by mucinous and signet ring cell histology compared to LOCRC (15.7% vs. 11.5% and 3.8% vs. 0.8%, respectively; *p*< 0.001) [93].

In conclusion, as shown in Table 2, young CRC patients usually have a good performance status and fewer comorbidities compared to older patients. However, they often are diagnosed at a metastatic stage and show worse prognostic features.

## 6. Molecular Signatures

Recent studies hint at a substantial molecular difference between EOCRC and LOCRC [94,95,96]. Hereditary cancer predisposing syndromes caused by tumor mutations account for approximately 30% of EOCRC diagnosis, whereas the remaining 70% have familial (20%) or sporadic (50%) forms, which represent a challenging topic for research [20]. Despite the increase in EOCRC incidence, however, there are still few data on the molecular features that lead to tumor initiation and progression in this population [97]. Genetic factors certainly play a key role in the occurrence of EOCRC, and it should be considered that having a negative family history does not exclude the presence of hereditary syndromes [31].

The three main mechanisms underlying the occurrence of CRC are as follows [9,20,96]:(1)Chromosomal instability (CIN), which is the classic pathway and causes mutations of multiple oncogenes, such as KRAS, and deletions of suppressor genes, such as APC and p53;(2)Microsatellite instability (MSI), which is usually sporadic, but there are also hereditary causes. MSI-CRC can be caused by hereditary germline mutations occurring in the MMR genes (MLH1, MSH2, MSH6, and PMS2) and tumor somatic hypermethylation of MLH1. Nearly 10% of CRCs are caused by MLH1 promoter hypermethylation and are often in association with the BRAF V600E pathogenetic variant in sporadic CRC [98];(3)CpG island methylator phenotype (CIMP) pathways. In some cases, CRC may be caused by the hypermethylation of promoter CpG island sites, resulting in the silencing of suppressor genes. Sometimes, there is MLH1 gene involvement and, thus, microsatellite instability [99,100,101].

In addition, according to some studies, inflammation may also play a role in the occurrence of sporadic EOCRC [102]. For example, a pathway involving TNF-R1 has been identified. TNF-R1 mediates inflammation, regulates cell proliferation, differentiation, and apoptosis [96], and promotes colon tumor initiation in IBD, independently of chronic inflammatory disease; however, it is still unknown what causes TNF-R1 activation in young patients [103,104]. Moreover, some evidence shows that a deficiency in NRF2 activity may promote EOCRC, as it causes oxidative stress, an accumulation of reactive oxygen species, and inflammation [89]. Finally, in a study of patients with IBD and EOCRC, a low prevalence of mutations in APC and PIK3CA and a high rate of mutations in TP53 were noted [8]. However, it is important to clarify that EOCRC and IBD are two distinct entities from a biological, clinical, and molecular point of view [8].

### 6.1. CIN

Few studies in the literature have investigated mutational rates in EOCRC. Mechanisms underlying CIN differ in EOCRC and LOCRC. Late-onset CIN tumors often lose chromosomal loci containing SMAD4, DCC and APC genes, whereas early onset CIN tumors gain chromosomal loci with BMPR1A and AMP-kinase regulatory subunits and lose loci containing FOX transcription factors and TJP2, a CRC marker [31].

The CIN pathway is followed by about 85% of CRCs. The primum movens is the loss of function mutations in the APC gene involved in the WNT/β-catenin signaling pathway, which leads to the formation of aberrant crypt foci. This leads to mutations in KRAS and TP53 and, thus, to the transformation of the adenoma into carcinoma [16].

The number of amplifications and deletions does not correlate with age. On the other hand, there is no clear relationships between patient age and global methylation levels, copy number variation (CNV) events, or a correlation with genomic clusters [26].

The literature data on the mutational incidences of genes are controversial, perhaps because older studies do not clearly distinguish between MSS and MSI tumors; in particular, there are many discrepancies for KRAS and BRAF V600E mutations [96]. Some studies report that EOCRC has a lower rate of mutations in BRAF V600E, NRAS, and KRAS; mutations in these genes appear to increase with age [8,16,96,105,106]. Stoffel and Murphy compared EOCRC and LOCRC, showing a lower prevalence of APC and BRAF somatic mutations in the young compared to the elderly and, at the same time, showing epigenetic changes indicative of the global hypomethylation of DNA than LOCRC [89]. Other studies, however, report a higher prevalence of KRAS mutations in young people than in the elderly (54% vs. 40%). The same applies to BRAF V600E mutations [31]. BRAF-mutated tumors are more frequently CIMP-positive compared to BRAF wild-type ones, more often involve the left colon, present more frequently with Stage IV diseases, and are less differentiated [105]. There appears to be no gender differences in BRAF mutation rates [105]. Finally, while in LOCRC patients, the BRAF mutation is often associated with MSI-H, all BRAF-mutated EOCRCs are MSI-L [105].

Moreover, there appears to be no difference between young and old patients regarding mutations on codons 12 and 13 of KRAS. The percentage of tumors with p53 overexpression seems to be lower in EOCRC than in LOCRC [9], but according to other studies, the mutation of P53, such as that of APC and PIK3CA, seems to be higher in young people.

Few studies in the literature focused on metastatic disease. Ting Xu et al. analyzed mutational differences in various genes between young and elderly metastatic patients, including KRAS (50.3% in mEOCRC versus 56.4% in mLOCRC, *p* = 0.32), NRAS (6.1% in mEOCRC versus 5.5% in mLOCRC, *p* = 1), BRAF (10.9% in mEOCRC versus 4.5% in mLOCRC, *p* = 0.056), PIK3CA (13.3% in EOCRC versus 12.7% in LOCRC, *p* = 1), and BRAF V600E (8.8% in mEOCRC versus 1.8% in mLOCRC, *p* < 0.01). Of note, mutations in the Wnt pathway were most common in LOCRC patients compared with EOCRC patients (100% versus 69.4%, *p* < 0.0001), whereas the P53 pathway alteration involved 85.2% of EOCRC patients and 81.2% of LOCRC patients (*p* = 0.499). Finally, no significant differences were detected between EOCRC and LOCRC patients in the mutation frequencies of the PI3K, MAPK, or TGF-b pathways [107] (Table 3).

Furthermore, another study observed mutational differences in patients with mCRC. BRAF mutations were rare. Conversely, KRAS exon 2 mutations showed high prevalence in mEOCRC; therefore, a relatively high KRAS mutation rate might indicate that this alteration influences biological behavior in mEOCRC [70].

In conclusion, much still needs to be understood about the molecular profile of mEOCRC, and further studies are needed to understand whether this subgroup of tumors can benefit from specific target therapies, in particular anti-EGFR and anti-VEGF monoclonal antibodies and whether predictive somatic markers are present. In addition, certain mutations seem to predict prognosis and deserve further attention. One study showed that mutations involving an increased expression of CK20, MAP3K8, EIF5A, and IMPDH2 are linked to a worse prognosis [97,108].

### 6.2. MSI and MSS

MSI is not exclusive to HNPCC. Although MSI is found in more than 90% of hereditary nonpolyposis colorectal cancer syndrome (HNPCC) patients, it is also present in 10% to 20% of sporadic CRC patients [4,109]. 

Germline mutations in the DNA MMR genes hMLH1 and hMSH2 and, to a lesser extent, hMSH6 and hPMS2 are associated with an increased lifetime risk of CRC and extracolonic cancers [110]. Although EOCRC is a marker of a potential hereditary component, the proportion of MSI tumors found within EOCRC varies from 19.7% to 41.0% [97].

MSI-H tumors more frequently involve the right colon, are mucinous, have normal p53 expression, have a higher frequency of metachronous and synchronous tumors, and have a better prognosis than MSS [5,9]. No gender differences appear to be present, although some studies have found a higher rate of MSI in women [109].

MSS tumors have less involvement of the right colon and fewer synchronous and/or metachronous tumors [5]. Moreover, there are differences between MSS tumors in young and elderly patients; in fact, in young patients, MSS tumors involve more distal sections of the colon, have a lower rate than other primary neoplasms, and show an important family component [5].

Up to 63% of EOCRC with MSS is euploid (CIN-negative) [95,111,112,113,114]. This type of cancer involves the rectum in 60% of cases (in contrast to MSI, which has a rate of around 19%), and patients often have a positive family history of cancer, which suggests genetic transmissibility [31]. 

The data on the frequency of MSI in the young and old are contradictory in the literature. Some studies argue that most EOCRC is MSS. MSI-h does not appear to be the main carcinogenic pathway [5,91], and in 80% of EOCRC cases, there is stability of the microsatellites [31,91]. This means that only 20% are unstable [31], and of these, almost all have Lynch syndrome. Furthermore, most sporadic colon cancers in young people are not caused by hereditary syndromes and are not attributable to germline mutations [89]. Conversely, in other studies, MSI appears to have a higher frequency in the young than in the elderly [4,9]. A study found a higher proportion of MSI cases in the younger (22.9%) than in the older cohort (20.6%) [4].

Differences between the young and old are also present when considering MSI-H tumors; in fact, BRAF is mutated in 5% of EOCRC MSI-H cases and in 48% of LOCRC cases [89]. The BRAF V600E mutation is linked to a methylation of the MLH1 promoter, and this causes microsatellite instability. In young patients, MSI seems to be mostly caused by Lynch syndrome, which may explain the absence of the coexistence of MSI-H and BRAF V600E mutations in EOCRC [107].

Lynch syndrome-related genes are more frequently mutated in the young than in the old, mostly with differences in MSH2 (2.7% vs. 0%; *p* = 0.004) and in MSH6 (4.8 vs. 1.2%; *p* = 0.005). These data suggest that younger CRC patients may be more likely to benefit from immunotherapy with checkpoint inhibition [115]. 

In another study, losses of MLH1 and PMS2 were more frequent among the elderly [30]. The loss of MLH1 is usually observed in LOCRC, although in some studies, it was also frequently observed in EOCRC [109]. Furthermore, isolated losses of hMSH2 without the loss of hMSH6 were found in both EOCRC and LOCRC [109].

The inactivation of MMR genes in young patients without a family history of colon cancer may be caused by de novo germline mutations or by two somatic mutations [9]. Epigenetic changes, which are generally related to environmental influences, appear to cause inactivation of the MLH1 gene [9,116]. In a study conducted in Taiwan, a Western-style diet, which is more prevalent in the young than in the elderly, appears to cause genetic or epigenetic alterations that lead to MSI. A Western diet and lifestyle expose young people’s MMR genes to yet unknown environmental factors that cause genetic and epigenetic alterations, thus leading to the MSI-H phenotype [9].

Regarding the metastatic stage, young patients seem to have higher rates of dMMR compared to their older counterparts. As is well known, dMMR is a positive prognostic factor, but in some studies, it seems that this applies only to Stages II and III. In fact, in one study, it was observed that young patients (aged under 30 years) with metastatic colon cancer and dMMR have poorer survival than pMMR patients [11]. 

Moreover, Ting Xu et al. evaluated 31 patients with mEOCRC, among which 16 were MSS and the remaining 15 were MSI-H. Sixteen patients had mutations in one of the MMR genes (six in MLH1, three in MSH2, two in PMS2, and five in MSH6). In addition, germline mutations were found in MUTYH (12% of patients), RAD50 (9%), TP53 (3%), and FANCL (3%). No patients showed mutations in APC or POLD1/POLE [107]. 

### 6.3. MACS (Microsatellite and Chromosome Stable)

MACSs are more frequent in young people, are often already metastatic at diagnosis, and have a worse prognosis than CIN and MSI [5,20]. However, there is still little information on MACSs, especially from a molecular point of view, but they seem to have a different methylation pattern involving genes that are still poorly understood. They are also associated with CIMP-low, rarely have BRAF mutations, and seem to show a different hypomethylation pattern than MSI and CIN CRC [5,20].

### 6.4. Molecular Differences Depending on Primary Tumor Location

Conversely to the traditional anatomical distinction, the colon cannot simply be divided into the right and left but must be considered a continuum of molecular alterations that fade into each other upon progression in the colic tract [117]. There is still little information on the molecular characteristics of the various colon localizations [5]. Whereas in EOCRC, right-sided colon tumors displayed a better prognosis, LOCRC showed a longer survival in left-sided tumors [118]. 

In EOCRC, MSI is predominantly in the right colon compared to the left colon (30% vs. 17%) [118]. In the ascending colon, MSI is caused by BRAF mutations and/or hypermethylation of the MLH1 promoter; in these cases, we speak of CIMP-high-component [119]. Conversely, in LOCRC, right colon tumors show BRAF mutations. 

Right colon tumors have a higher rate of BRAF mutations than left ones [120]. Furthermore, CIMP and MSI mainly involve the proximal colon, unlike P53 and CIN that involve the distal colon and rectum [5]. Recent results from a cohort of left-sided and rectal EOCRC showed higher mutation rates of NF1, POLE, SMAD4, and BRCA2. Furthermore, mutations in the left-sided EOCRC genes involved in histone modification, such as KDM5C, KMT2A, KMT2C, KMT2D, and SET2D, were reported more frequently [20] (Figure 4).

Surely science will have to strive to understand in detail what differences are present in the right and left colon between the young and the old, so that oncologists can be better guided in their choice of treatments.

## 7. Treatment

To date, an early age of onset does not influence the treatment algorithm for CRC in the major clinical guidelines, and EOCRC patients have the same therapeutic indications as their older counterparts. Nevertheless, in clinical practice, the age at diagnosis might influence physicians’ decision-making. Considering that younger patients tend to have better performance status and fewer comorbidities, the evidence shows that EOCRC is likely to receive more aggressive treatment, with conflicting data on the potential prognostic advantages [81,121,122]. An American cohort study involving 13,102 EOCRC patients (18 to 49 years) and 37,007 LOCRC patients (65 to 75 years) treated with curative-intent surgery and adjuvant chemotherapy reported that Stage IV CRC patients younger than 40 tended to undergo more systemic therapy and radiation with minimal survival gain (relative risk 0.84; 95% CI 0.79–0.90) [81]. These data were also confirmed in a Canadian retrospective study [121]. A recent analysis of 554 mCRC patients showed that EOCRC patients (≤50 years) received more lines of chemotherapy (2.94 vs. 2.38, *p* = 0.027) and underwent more surgeries (2.42 vs. 1.24; *p* < 0.001) than their older counterparts. Furthermore, the younger cohort demonstrated better survival in first-line treatment (mPFS 16.2 vs. 11.3 months, *p* = 0.042; mOS 121.5 vs. 58.1 months, *p* = 0.011) [122]. Other data suggested no significant difference in the recurrence rate in radically resected Stage III and IV young patients (27.5% vs. 27.9%, *p* = 0.325), even though they had more advanced stages at diagnosis (55.6% vs. 47.9%, *p* = 0.001) and worse cancer-specific survival (81.2% vs. 87.8%, *p* < 0.001) [123]. In another American retrospective analysis, a more intense care pattern (more metastatic resections and triplet chemotherapy) was seen in younger patients; however, age did not predict OS [80].

The only potentially curative option for mCRC is radical surgery, which should always be performed when feasible. Liver metastasis resection should be considered when sufficient remnant parenchyma is preserved (≥30% liver). Loco-regional ablative treatments, such as radiotherapy, thermal ablation, and chemoembolization, might be associated with surgery. However, up to 55–80% of patients will experience relapse following metastatic resection [124]. Cytoreductive neoadjuvant combination chemotherapy (doublet or triplet) with or without a targeted agent might be considered in fit patients with potentially resectable mCRC [125]. A retrospective analysis of 573 patients who underwent resection of CRC liver metastases reported a greater negative impact on survival of RAS mutations in the EOCRC (HR 2.03, 95% CI 1.30–3.17) compared to the LOCRC cohort (HR 1.64, 95% CI 1.23–2.20), with an even worse impact in patients younger than 40 (HR 2.97, 95% CI 1.44–6.14) [126].

In order to optimize the treatment choice, testing for KRAS, NRAS exon 2, 3 and 4, BRAF mutations, and MMR status should be carried out in every new metastatic CRC diagnosis [124]. RAS mutations are predictive of resistance to EGFR inhibitors [127,128,129], while the BRAF V600E mutation represents a negative prognostic factor [130]. Among RAS wild-type patients, only left-side CRC benefits from the combination of chemotherapy and anti-EGFR, whereas right-sidedness predicts poor prognosis and limited response to these agents [131]. Right-sided RAS wild-type patients might be selected for a doublet chemotherapy with an anti-EGFR only when a good response is needed (e.g., for conversion therapy) [124]. A post hoc analysis of the Valentino trial—the phase II randomized study assessing the efficacy of maintenance therapy with panitumumab alone or plus 5-FU/LV in RAS wild-type mCRC—hinted that age did not influence survival but determined a different toxicity profile, with increased anemia and a skin rash and decreased hypomagnesemia in patients younger than 50 [132].

Palliative fluoropyrimidine-based backbone chemotherapy, combined with anti-VEGFR bevacizumab, represents the standard first-line treatment for metastatic, proficient MMR, RAS, or BRAF-mutated CRC. Patients deemed unfit for combination treatments might receive single-agent fluoropyrimidine with or without bevacizumab [124]. Patients with an adequate performance status should receive doublet chemotherapy with oxaliplatin or irinotecan or the triplet FOLFOXIRI (5FU, LV, oxaliplatin, and irinotecan), which has demonstrated increased tumor shrinkage (RR 66% versus 41%; *p* = 0.0002) and survival (median PFS 9.8 versus 6.9 months; HR 0.63; *p* = 0.0006; median OS 22.6 versus 16.7 months; HR 0.70; *p* = 0.032) compared with FOLFIRI (5FU, LV, and irinotecan) at the price of a worse toxicity profile [133]. BRAFV600E-mutant tumors showed no benefit from the triplet plus bevacizumab; therefore, FOLFOX with bevacizumab represents the standard-of-care first-line treatment in this population [134]). Nevertheless, FOLFOXIRI with or without bevacizumab remains a viable option in right-sided BRAF-mutant mCRC patients [135]. Based on a retrospective analysis involving 196 patients with treatment-naïve right-sided mCRC, younger patients received triplet regimens more often but underwent fewer subsequent therapies (61% versus 78%, *p* = 0.043) [136]. Pooled analysis of the TRIBE and TRIBE2 studies demonstrated that intensification of upfront backbone chemotherapy in younger patients (<50 years) had a lower risk of G3-4 neutropenia, diarrhea, and asthenia and a higher risk of nausea and vomiting, with no significant survival benefit difference between the two age cohorts [137]. Therefore, considering the positive prognostic implications of the continuum of care, it is crucial to underline the importance of careful patient selection for the intensification of treatment, which covers its primary role when a maximal tumor shrinkage is needed. 

The MMR status aids clinicians to identify patients who might benefit from first-line immune checkpoint inhibition [138,139] and to select those who should undergo genetic counseling for the identification of Lynch syndrome. Full germline genetic testing should be offered to every new MSI-H/dMMR cancer diagnosis, excluding those who display the BRAF V600E somatic mutation and/or hypermethylation of the MLH1 promoter, and families with high clinical risk [134,135].

Patients with good PS and no significant comorbidities who progress on a first-line therapy should be offered a second-line treatment. The management is driven by the previous regimen administered: the switch from a first-line oxaliplatin-based regimen to a second-line irinotecan-based regimen and vice versa represents the standard option. Antiangiogenic and anti-EGFR antibodies might be added to backbone chemotherapy, according to the presence or absence of RAS mutations [124]. Furthermore, data from the phase III BEACON CRC trial showed a survival benefit from the combination of encorafenib (BRAF inhibitor) and cetuximab (EGFR inhibitor) for pretreated BRAF V600E-mutant patients [140].

Patients fit for beyond second-line treatment might receive oral agents, such as regorafenib and trifluridine–tipiracil (TAS-102), which demonstrated increased survival compared to BSC in phase III trials [141]. The phase III SUNLIGHT trial recently demonstrated a significant survival gain from TAS-102 plus bevacizumab compared with TAS-102 alone (median OS 10.8 months versus 7.5 months; HR 0.61; *p* < 0.001) [142] and a manageable safety profile in mCRC treated with 1–2 prior chemotherapy lines. Furthermore, enrollment in clinical trials with anti-HER agents in treatment-refractory HER2-positive mCRC is strongly encouraged [143]. Patients with no satisfactory treatment options expressing a neurotrophic tyrosine receptor kinase (NTRK) gene fusion are candidates to receive the tyrosine kinase inhibitors entrectinib [144] and larotrectinib [145].

Much work is underway to identify other potentially targetable driver mutations for EOCRC in order to design personalized therapeutic approaches and minimize chemotherapy-related side effects. A comprehensive analysis of 18,218 specimens collected from younger (<40 years old) and older (≥50 years old) CRC patients demonstrated that overall genomic alterations were similar in most of the studied genes. Younger patients had frequent TP53 and CTNNB1 alterations, whereas APC, BRAF, FAM123B, KRAS, and PIK3CA mutations were observed mostly in older patients. Interestingly, TMB in the MSI-H cohort increased with age [146]. Genetic alterations that differ with age might provide promising insights for personalized treatments; therefore, further research in this field is needed.

To conclude, prognostic features, predictive biomarkers, and specific treatment algorithms for EOCRC are yet to be determined. To date, there is no evidence that the intensification of therapies confers survival benefits; therefore, a heavier toxicity profile and long-term implications should be considered in treatment planning for young mCRC patients.

## 8. Conclusions

The frequency of colon cancer is increasing in the population under the age of 50 years old compared with a decrease in cases in older people. The reasons behind this changing phenomenon are still unknown. Certainly, there is a multifactorial component, but scientific research efforts will have to be focused on understanding the pathogenesis of these cancers to counteract their spread.

As the younger population is not covered by screening programs, EOCRC is often diagnosed at a more advanced stage. Furthermore, mEOCRC shows different features compared with mLOCRC, being biologically more aggressive and less differentiated. From a clinical and therapeutic point of view, mEOCRC does not differ from its counterpart, although young people often tend to undergo more aggressive treatments due to their better performance status. From the molecular perspective, however, studies do not always concur, even though, in general, mEOCRC seems to have more KRAS, Wnt, and p53 mutations, and a higher rate of dMMR.

In conclusion, much effort still needs to be made to understand this emerging issue, starting from the identification of predisposing factors and the development of new preventive strategies. Given the alarmingly rising incidence of EOCRC, it might be useful to consider modifying the screening programs and including younger patients with no familial predisposition, even if this would have a significative impact on the healthcare economy. It might be possible to think of charging younger people for the testing of occult blood in stools, guaranteeing tax relief in any case, and performing only colonoscopies at the full expense of the healthcare system.

## Figures and Tables

**Figure 3 cancers-15-03509-f003:**
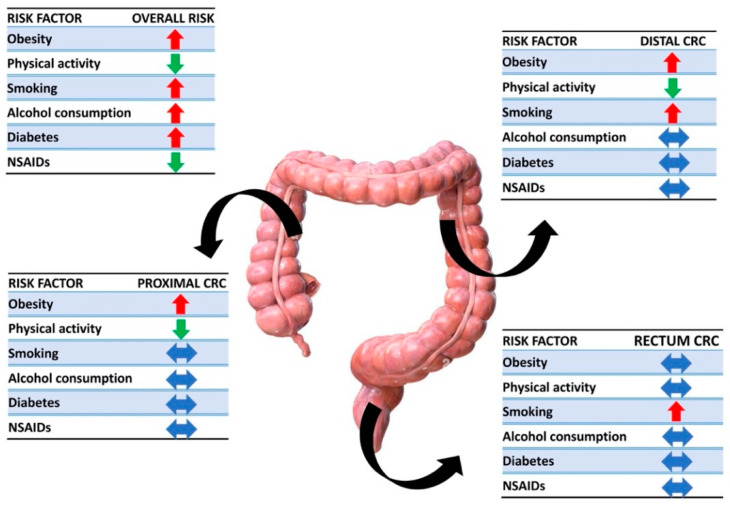
Associations between risk factors and anatomical sites. Red arrows are placed next to risk factors that have been shown to have a role in cancer promotion, green arrows identify protective elements, and blue arrows represent no correlations.

**Figure 4 cancers-15-03509-f004:**
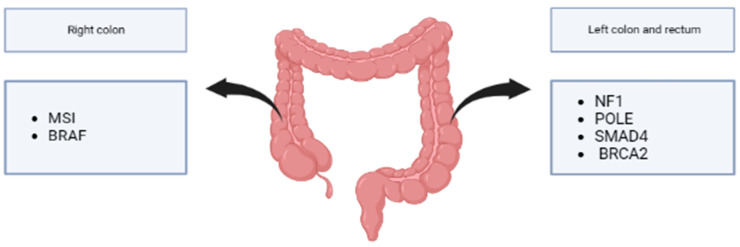
Mutation predominance in EOCRC in right vs. left colon. Right colon tumors have a higher rate of BRAF and MSI mutations than left colon, and left-sided and rectal showed higher mutation rates of NF1, POLE, SMAD4, and BRCA2.

**Table 1 cancers-15-03509-t001:** Hereditary and nonhereditary risk factors associated with EOCRC development.

Risk Factors	
Hereditary factors	Lynch syndrome
	Adenomatous polyposis syndromes
	Li-Fraumeni syndrome
	Cystic fibrosis
Nonhereditary risk factors	Obesity
	Type II diabetes mellitus
	Western diet
	Being sedentary
	Alcohol and tobacco use
	Chronic inflammation
	Intestinal dysbiosis
	Pollution

**Table 2 cancers-15-03509-t002:** Differences between EOCRC and LOCRC [5,29,84,85,86,91,92].

Differences	EOCRC	LOCRC
Stage at diagnosis	Advanced (III–IV)	Less advanced
Tumor location	Left	Right
Risk of synchronous or metachronous tumors	Higher	Lower
Histology	Mucinous and signet ring cell	
Histopathological features	Perineural, lymphovascular, and venous invasion	

**Table 3 cancers-15-03509-t003:** This table summarizes the differences in the frequency of different mutations. PIK3CA and P53 are more frequent in mEOCRC; conversely, Wnt, APC, SMAD4, and DCC are mainly in mLOCRC. There are no differences in mutations at codons 12 and 13 of KRAS. Finally, studies report discordant data for NRAS, KRAS, BRAF V600E, and BRAF.

	mEOCRC	mLOCRC
PIK3CA	↑	
P53	↑	
Wnt		↑
APC		↑
SMAD4		↑
DCC		↑
Codons 12 and 13 of KRAS	=	=
NRAS	?	?
KRAS	?	?
BRAF V600E	?	?
BRAF	?	?

## Data Availability

The data can be shared up on request.

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
