# Peer review of "Early Onset Metastatic Colorectal Cancer: Current Insights and Clinical Management of a Rising Condition"

_cancers, 2023, doi:10.3390/cancers15133509_

Round 1

Reviewer 1 Report

Overall the paper is comprehensive review on the data for early onset colorectal cancer. The global outlook is particularly laudatory. The paper would benefit from some additional tables and figures to accompany some of the data as the material can become dense. This will help the flow of the manuscript overall. Areas to add figures include

a. incidence rate data

b. risk factors description

c. table is provided for mEOCRC mutational differences for one paper by Ting Xu et al but consider a broader table of all of the other studies

additional edits include

a. describe all the genes involved in APS (adenomatous polyposis syndrome)

b. describe risk factor for EOCRC rectal cancers since all of the factors appeared to be for colon cancer

c. On page 8, please describe CIMP pathways. No description is given

In addition, editing of the manuscript for brevity and flow will also help the manuscript.

minor edits needed

Author Response

Thank you for your kind reminders and the interest you showed in our work.

As you suggested, we added some additional tables and figures regarding:

  1. incidence rate data
  2. risk factors description
  3. a broader table for mEOCRC mutational differences

Furthermore, we included a description of all the genes involved in APS (adenomatous polyposis syndrome) and added some reference to risk factors for EOCRC rectal cancers, especially a table regarding the association between risk factors and anatomical sites.

On page 10, we added a part to anticipate CIMP pathway, better described during the argumentation of molecular characteristics of EOCRC.

In addition, we edited the manuscript for brevity and flow in order to make it more readable.

Reviewer 2 Report

The authors present and excelent manuscript. It is a review of the actual state of knowledge about Early onset metastatic colorectal Cancer. The authors present an excellent review on "Early onset metastatic colorectal cancer: current insights and clinical management of a rising condition" The study is well design and comprehensively presented. The main question addressed by the research is the analysis of the actual siituation of Colorectal cancer in young people (under 45): epidemiology, pathophyology, diagnosis, treatment and prognosis. The objective is well defined and described. Objectives are well described. The backgraunds of the problem are well reviewed and described. The epidemiological information is are very robust. - The topic is relevant in the field of CRC. CRC in young people is a rising problem , due to an incresed incidence, dificult diagnosis and poor prognosis. - The manuscript is written in a very clear and comprehensive style. The review is very thorough and up-to-date. - One specific improvement respecting other published material is the sound and detailed review of the role of genetic alterations in the genesis and progression of CRC in young people and the differences respecting old patients. The paper´s structure flows smoothly and is logically well written and very informative. The Tables are informative and helpful. References are appropiate. The revision of the literature is thorough.

Author Response

Thank you for your comments. We are pleased that you appreciated our work and we hope that this might help other medical professionals in facing this alarmingly rising condition.
